# Geometric Deep Learning and Heatmap Prediction for Large Deformation Registration of Abdominal and Thoracic CT

**In Young Ha**[1]                                    HA@IMI.UNI-LUEBECK.DE

**Lasse Hansen**[1] **Matthias Wilms**[2] **Mattias P. Heinrich**[1]  HEINRICH@IMI.UNI-LUEBECK.DE

[1] *Institute of Medical Informatics, University of Luebeck*

[2] *Department of Radiology, University of Calgary, Calgary, Canada*

## Abstract

We propose a novel concept for supervised learning of image registration for large deformations. Based on ideas from discrete graphical models in image registration, we design a network architecture that learns to predict discrete heatmaps for the relative displacement of a number of sparse keypoints between two scans. Graph convolutions are used to model a globally smooth transformation and deformable convolutions are used to learn suitable features representations and a similarity metric to estimate sparse displacements based on the volumetric scans in an end-to-end manner. Experimental validation for weakly-supervised label-driven registration demonstrates an improvement of 10% points of overlap accuracy compared to a state-of-the-art deep learning approach.

## 1. Introduction

A large number of recent works have addressed medical image registration with deep convolutional networks (Balakrishnan et al., 2019; Hu et al., 2018), which enable rapid inference times and the potential of optimising challenging alignment tasks with regards to expert supervision labels. Yet, large and complex deformations remain problematic for classical feed-forward regression networks, which led to errors of >2.5 mm for inhale-exhale lung CT registration in (Eppenhof et al., 2018; de Vos et al., 2019; Sentker et al., 2018), which is inferior to conventional methods with errors of less than 1 mm (Rühaak et al., 2017).

Heatmap regression is often used for human joint detection (Bulat and Tzimiropoulos, 2016) and enables a probabilistic estimate of global keypoint locations, which can be extended to the regression of relative displacement vectors for sparse correspondences. Graphical models have excelled at estimating pairwise correspondences using densely quantised local displacement windows (Glocker et al., 2008; Heinrich et al., 2015).

## 2. Methods

Our proposed deep geometric registration framework is trained with discrete heatmaps derived from weak correspondences of segmentations evalutated for sparse keypoint locations.

1. a convolutional feature extraction part that learns a mapping from the input intensities to 16-dimensional feature descriptors;

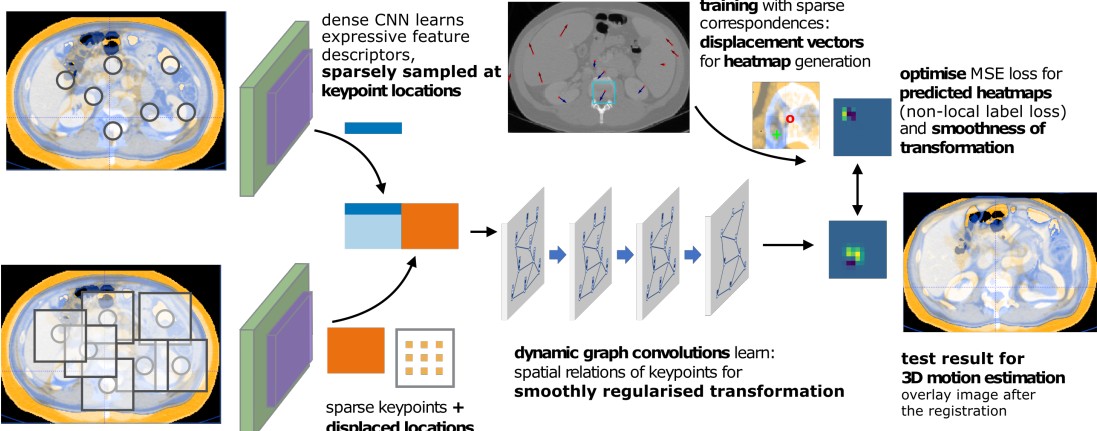

Figure 1: Concept of proposed geometric deep learning architecture for discrete estimation of large displacements. A CNN extracts low-level 16-dim. features from images, which are sparsely sampled at keypoint locations in the fixed scan. A quantised grid of displacement is considered around each control point for the moving scan, e.g. using 13×13×13 offsets. For the graph convolution, fixed and moving features are concatenated for each keypoint and each displacement. We learn to predict displacements that yield a globally smooth transform and are locally similar to the ground truth ones.

2. a set of keypoints that are first used to define control points for which the local similarity over all considered displacements between fixed and moving features is calculated;

3. a graph convolutional network based on a kNN-graph (between keypoints) that regularising the transformation globally using graphical message passing; and

4. two loss terms that optimise the accuracy of the predicted displacement heatmaps (non-local label loss) and the regularity of the regressed 3D displacement field based on the weighted kNN-graph using a mean-squared error.

These steps are trained first individually for a sufficiently good initialisation and then trained in an end-to-end fashion. Many different algorithmic choices are possible within this presented framework and we restricted our initial experiments to Obelisk feature extractors (Heinrich et al., 2019) and dynamic graph CNN (Wang et al., 2018). To extend the concept of heatmap regression from landmark localisation to **weakly label-driven supervision** for sparse correspondence search, we compute *pseudo heatmaps* based on one-hot representations of segmentation labels.

## 3. Results and Conclusion

Quantitative experiments were performed for a three-fold cross-validation for 10 scans of the VISCERAL anatomy 3 dataset (Jimenez-del Toro et al., 2016) with 9 anatomical segmentation labels: ■ liver, ■ spleen, ■ pancreas, ■ gallbladder, ■ bladder, ■ r. kidney, ■ l. kidney, ■ r. psoas muscle (psoas) and ■ l. psoas. We consider the task of atlas-based **align-**

Table 1: Quantitative cross-validation for 10 scans of the VISCERAL anatomy 3 dataset. The initial alignment for the 9 labels was 36.1%.

| Method | 🟥 | 🟩 | 🟦 | 🟨 | 🟦 | 🟪 | 🟥 | 🟧 | 🟩 | ∅ (avg of labels) |
|---|---|---|---|---|---|---|---|---|---|---|
| Label-Reg | 75.7 | 50.8 | 14.7 | 7.8 | 48.6 | 59.1 | 60.0 | 55.5 | 59.1 | 47.9% |
| corrField | 73.2 | 67.1 | 17.7 | 5.9 | 48.5 | 54.3 | 70.9 | 47.2 | 59.7 | 49.4% |
| SSC+diff.reg | 69.4 | 38.9 | 0.0 | 4.1 | 51.2 | 39.5 | 47.2 | 48.0 | 51.4 | 38.9% |
| **proposed** | 83.1 | 65.0 | 25.2 | 22.3 | 54.9 | 74.6 | 75.8 | 66.5 | 65.5 | **59.2%** |

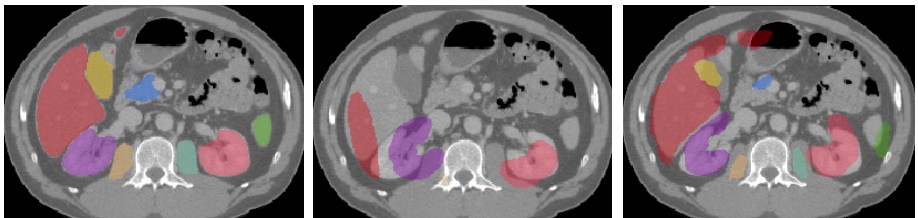

Figure 2: Left: CT scan with ground truth labels. Middle: Same CT with segmentation of moving scan. Right: Visual outcome of our proposed deep geometric heatmap registration used to transfer the moving segmentation to match the target scan.

**ment of abdominal anatomies** across patients with large deformations. That means the keypoints of one scan are sampled based on the class distribution of segmented organs, while this information is unknown for the second scan. We compare our proposed approach to **Label-Reg**, a label-driven deep learning registration tool (Hu et al., 2018) and **corrField**, an MRF-based conventional registration method for keypoints (Heinrich et al., 2015). A dense deformation field was estimated for our method using a least-squares fitting based on the sparse displacement probabilities. Table 1 reports the average Dice scores across 66 registrations with a quantitative gain of approx. 10% points Dice for our method. A visual example is shown in Fig. 2. The average 3D inference time is around 3 sec.

**Conclusion:** We have presented a new concept for large deformation estimation using graph convolutions and discrete regression of displacement heatmaps. The initial results for abdominal CT are very encouraging, outperforming previous approaches by a large margin. Thoracic lung registration is considered in our ongoing research, which also includes model-based augmentation to deal with the small number of training scans (Uzunova et al., 2017).

## Acknowledgments

This work is funded by the German Research Foundation DFG (HE 7364/2).

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
