# OpenReview forum: "Geometric Deep Learning and Heatmap Prediction for Large Deformation Registration of Abdominal and Thoracic CT"
_MIDL.io/2019/Conference/Abstract — MIDL Abstract 2019_

### Official Review · AnonReviewer2 · 2019-04-28

**Rating:** 3
**Confidence:** 1

**Review:**

A general deep learning based registration framework is proposed which would find its use in various large deformation registration tasks. The framework accomodates different CNN keypoint feature extractors and graph CNNs.
Method is well validated. Compared to two baselines and it substantially outperforms both of them.

---

### Official Review · AnonReviewer1 · 2019-05-01
**A good paper on deformable registration with deep learning**

**Rating:** 3
**Confidence:** 2

**Review:**

The authors propose a deformable registration method based on deep learning. They use graph convolutions (based on a KNN-graph on the keypoints or landmarks) and a heat map regression. One of the advantages of this approach is that it enables to perform registration when large deformations exist. The methodology is sound and the results seem to be competitive. I have several small concerns though. First, it would be interesting (if possible) to measure the error/success based on the deformation complexity. For example, I would expect the method to yield results similar to other competing methods when the deformations are small, and to outperform the others when these deformations become large. Second, is Label-Reg the state-of-the-art in registration? And third, it is not clear to me how segmentation (for example middle image on Figure 2) is achieved.

---

### Decision · Program_Chairs · 2019-05-06
**Acceptance Decision**

Accept